# Temperature Compensation of the MEMS-Based Electrochemical Seismic Sensors

**DOI:** 10.3390/mi12040387

**Published:** 2021-04-02

**Authors:** Chao Xu, Junbo Wang, Deyong Chen, Jian Chen, Wenjie Qi, Bowen Liu, Tian Liang, Xu She

**Affiliations:** 1State Key Laboratory of Transducer Technology, Aerospace Information Research Institute, Chinese Academy of Sciences, Beijing 100190, China; xuchao16@mails.ucas.ac.cn (C.X.); chenjian@mail.ie.ac.cn (J.C.); qiwenjie16@mails.ucas.ac.cn (W.Q.); liubowen17@mails.ucas.ac.cn (B.L.); liangtian18@mails.ucas.ac.cn (T.L.); shexu18@mails.ucas.ac.cn (X.S.); 2School of Electronic, Electrical and Communication Engineering, University of Chinese Academy of Sciences, Beijing 100049, China

**Keywords:** temperature compensation, electrochemical seismic sensor, MEMS, thermistor

## Abstract

Electrochemical seismic sensors that employ liquid as their inertial masses have the advantages of high performances in the low-frequency domain and a large working inclination. However, the surrounding temperature changes have serious impacts on the sensitivities of the sensors, which makes them unable to work as expected. This paper studied the temperature characteristics of electrochemical seismic sensors based on MEMS (micro–electro–mechanical systems), and analyzed the influences of the temperature effects on the open-loop and closed-loop amplitude-frequency curves. Most importantly, the temperature compensation circuits based on thermistors were developed, which effectively adjusted pole frequencies and sensitivity coefficients, and finally realized the real-time temperature compensation for both open-loop and closed-loop measurements for the first time. The results showed that in the temperature range of −10 °C ~ +40 °C, and with the 3 dB bandwidth range of 0.01 Hz ~ 40 Hz, the change of the maximum sensitivity was reduced from about 25 dB before temperature compensation to less than 2 dB after temperature compensation.

## 1. Introduction

Seismic sensors are used to detect small and low-frequency ground vibrations, and they are widely used for natural seismic observations [1], geological structure surveys [2], underground oil and gas resource explorations [3], underwater target monitoring [4], and nuclear monitoring [5]. The electrochemical seismic sensor is a new type of geophone that uses liquid as the inertial mass and electrochemical reactions as the mechanism for energy conversion. Compared to other mechanical seismic sensors, the electrochemical seismic sensor has high sensitivities in the low-frequency domain, and a large working inclination [6].

The performance of commercial seismic sensors based on various principles are listed in Table 1. Compared to these two common seismic sensors, electrochemical seismic sensors can reach the same level of sensitivity and bandwidth as capacitive seismic sensors, and the working inclination is much larger due to the fluidity of the liquid mass than that of the capacitive type, which makes the electrochemical seismic sensors more suitable for applications in complex environments, such as applications in deep-sea. Additionally, the self-noise of electrochemical seismic sensors is a little worse than that of capacitive ones, due to the thermomechanical motion noise caused by the spring–mass vibrational system and the electrochemical process.

In addition, with the introduction of MEMS technologies, the performances of the sensitive electrodes of electrochemical seismic sensors were greatly improved [7,8,9,10,11], which attracted increasingly more interests from researchers all over the world.

However, the performances of electrochemical seismic sensors are greatly affected by the surrounding temperature. For the liquid environment, the convection and diffusion of ions in the electrolyte are affected by the temperature. In addition, the electrochemical reaction rate is also related to the temperature, and the elastic coefficient of the elastic membrane that provides elastic force is also seriously affected by the temperature. Thus, when the electrochemical seismic sensors work under terrible environments where the temperature changes greatly, the performances of the device deteriorates.

In order to study the temperature characteristics of electrochemical seismic sensors, Dmitry A. Chikishev from the Moscow Institute of Physics and Technology studied the influence of temperature on the amplitude–frequency characteristics of the sensor [13]. In their work, at a frequency range of 0.1 Hz ~ 443 Hz, and a temperature range of −15 °C/−35 °C ~ +70 °C, the open-loop sensitivity was tested. It was found that the sensitivity changed by orders of magnitudes with temperature. The data processing results showed that the values of characteristic frequencies of the open-loop sensitivity curves showed an exponential relationship with the reciprocal of temperature. They also verified the influence of temperature on the viscosity of the electrolyte. However, their work did not put forward an effective temperature compensation method, and the temperature characteristic for lower frequencies were not studied. Lin Jun from the Jilin University conducted studies of temperature compensation on the MTLS10 electrochemical seismic sensor [14]. Their study tested the sensitivity curves with a temperature range of 10 °C ~ 45 °C, and developed a temperature coefficient model, which corrected the temperature sensitivity through a mathematical model. However, this model had a poor correction effect below the frequency of 0.1 Hz, and the method could not realize real-time compensation.

In order to deeply understand the influence of temperature on the electrochemical seismic sensor and develop an effective method of real-time temperature compensation, this paper studied the temperature characteristics of wide-broadband electrochemical seismic sensors and managed to find an efficient method to realize real-time temperature compensation. The open-loop and closed-loop sensitivity curves of the sensors were tested with the frequency range of 0.01 Hz ~ 100 Hz and the temperature change of −10 °C ~ +40 °C, and transfer functions were established by typical elements. Additionally, the temperature compensation circuits were designed on the basis of thermistors to offset the changes in pole frequencies and sensitivity coefficients. Experimental results showed that the developed temperature compensation method could realize real-time temperature compensation, and the sensitivity changes of open-loop and closed-loop measurements were controlled within 2 dB.

## 2. Structures and Principles

### 2.1. Working Principles of the Elecrochemical Seismic Sensors

The structure of the electrochemical seismic sensor is shown in Figure 1a. The device was rigidly fixed on the ground surface to accurately sense ground motions. The structure included elastic membranes, liquid storage cavities, a compressed channel, sensitive electrodes, and a plexiglass shell. The elastic membranes were used to provide an elastic force, and the storage cavities were filled with electrolyte (I_2_&KI) and connected by the compressed channel. At the center of the compressed channel, there were two pairs of mesh-shaped sensitive electrodes that were used to sense the liquid flow. The sensitive electrodes were composed of two pairs of electrodes, used for differential output to eliminate the influence of common mode noises. Each pair of electrodes was divided into a cathode and an anode, and there were four electrodes in accordance with the anode–cathode–cathode–anode (ACCA) composition, see Figure 1c. The following electrochemical reactions occurred in the electrolyte after the electrodes were powered on.
(1)$$\begin{array}{l} \mathrm{anode}:3I^{-}-2e^{-}\to I_{3}^{-}\\ \mathrm{cathode}:I_{3}^{-}+2e^{-}\to 3I^{-} \end{array}$$

The whole electrolyte acted as a liquid inertial mass. When an external vibration occurred, the liquid inertial mass moved relative to the shell. Since the sensitive electrodes were fixed to the shell, the electrolyte near the electrodes changed its flow accordingly, which led to the change of the ion concentrations near the electrode surface. Finally, the corresponding current was obtained through the electrochemical reaction to characterize the amplitudes of external vibrations.

The whole transfer process could be divided into a vibration model and an electromechanical model. The vibration model referred to the process of converting the outside ground vibration into the flow velocity at the compressed channel, and the electromechanical model referred to the process of converting the flow velocity at the compressed channel into electrical signals.

For the vibration model, it could be equivalent to a typical second-order inertial damping system, as shown in Figure 1d. It consisted of a seismic mass *m*, attached to the measurement point through a suspension, represented by a spring *k* and a dashpot *c*. [15]

Under the ground displacement *x_g_*, the dynamic equation of the seismic mass was:(2)$$m\frac{d^{2}x_{m}}{dt^{2}}+c\frac{dy}{dt}+ky=0$$
where *x_m_* is the displacement of mass, *y* is the relevant displacement to ground, and *y* = *x_m_* − *x_g_*, thus, the Equation (2) could be rewritten as:(3)$$m\frac{d^{2}y}{dt^{2}}+c\frac{dy}{dt}+ky=-m\frac{d^{2}x_{g}}{dt}$$

By using the Laplace transform, the transfer function *H_m_*(*s*) of this typical inertial damping system could be expressed as Equation (4).
(4)$$H_{m}\left(s\right)=\frac{Y}{X_{g}}=-\frac{ms^{2}}{ms^{2}+cs+k}$$

For an electrochemical seismic sensor, the frequency response of the vibration model $\left|H_{mech}(\omega )\right|$ could be expressed as [16]:(5)$$\left|H_{mech}(\omega )\right|=\left|\frac{Y(\omega )}{X_{g}(\omega )}\right|=\frac{\omega^{2}}{\sqrt{{\left(\omega^{2}-\omega_{0}^{2}\right)}^{2}+R_{e}^{2}\omega^{2}}}$$

In which, ω0=Km is the natural frequency of the system, and *K* is the elastic coefficient of system, constituted by the membranes and the support spring at the bottom of Figure 1a; the movable coil made a lesser contribution to the spring constants. *m* is the liquid mass. *R_e_* is the equivalent flow resistance of the system. It indicated that the vibration model is the high-pass with the second-order.

The electromechanical model of the electrochemical seismic sensor could be described as the general mass transfer processes in liquid, and was expressed by the Nernst-Plank equation [17]. However, it is very difficult to obtain the analytical solution of the electromechanical model. The concentration and velocity distributions were needed to solve more complicated partial differential equations. In addition, it was also necessary to meet the complex electrochemical boundary conditions, such as the electrochemical law Butler–Volmer equation. In order to reduce the complexity, V.A. Kozlov from the Moscow Institute of Physics established the theory of the transfer function of electrochemical cells under convection–diffusion conditions in their early studies, and proposed the variational principle of the integration of the convection–diffusion equation. A simplified transfer function of the electromechanical model was proposed in (Equation (6)) [18]. In the equation, $\omega_{d}$ is the diffusion characteristic frequency, “~” indicates proportional to. However, this simplified module is difficult to accurately predict the degree of attenuation at high frequencies, only a trend analysis is available.
(6)$$\left|H_{elec}\left(\omega \right)\right|\sim \frac{1}{\sqrt{1+\omega^{2}/\omega_{d}^{2}}}$$

Then, the transfer function of the entire electrochemical seismic sensor could be expressed as Equation (7).
(7)$$\left|H\left(\omega \right)\right|=\left|H_{mech}(\omega )\right|\cdot \left|H_{elec}\left(\omega \right)\right|=\frac{\omega^{2}}{\sqrt{{\left(\omega^{2}-\omega_{0}^{2}\right)}^{2}+R_{e}^{2}\omega^{2}}}\cdot \frac{1}{\sqrt{1+\omega^{2}/\omega_{d}^{2}}}$$
where *ω*_0_ was quantified as 2.5 by the empirical data, and *w_d_* = *D*/*d*^2^; *D* was the diffusion coefficient of $I_{3}^{-}$, and was quantified as 0.574 × 10^−9^ m^2^/s, and d was the distance between the anode and the cathode. The distance was 10 μm in this study.

The fitting results are shown in Figure 2. The fitting results of high frequency were poor, but the trends were consistent.

### 2.2. The Feedback Processes

The sensitivity *S* [unit: V/(m/s)] of the velocity-type seismic sensors was defined as the ratio of the output voltage *U_O_* (unit: V) of the seismic sensor and the external vibration velocity *V_in_* (unit: m/s) at a certain vibration frequency *f*_0_ (unit: Hz). Thus, *S*(*f*_0_) = *U_O_*/*V_in_*. 

In actual application, the seismic sensors must ensure that the amplitude–frequency curve was flat, within the required bandwidth, so the concept of −3 dB bandwidth appeared. The −3 dB bandwidth referred to the widest frequency band with the attenuation of sensitivity in 3 dB (√2/2 times) rather than the highest sensitivity. Therefore, a feedback process was necessary to expand the −3 dB bandwidth. In order to express the −3 dB bandwidth conveniently, the values of sensitivity were processed logarithmically with 20 × log10(*S*) in this study, and the unit was dB.

The electrochemical seismic sensors employed electromagnetic force as their negative feedback to expand their −3 dB bandwidths [19,20], as shown in Figure 1a. The output currents of the seismic sensor passed through the signal pre-processing circuits (*H*_1_(*s*)) first, and was then processed by feedback compensation circuits (*H*_2_(*s*)) and temperature compensation circuits (*H*_3_(*s*)). After this, the adjusted signals were loaded on the coil through the feedback control circuits (*H_f_*(*s*)), and the coil was connected with the elastic membrane. Under the action of a fixed magnet, the coil together with the elastic membrane was subjected to electromagnetic force, which acted as the feedback velocity signal on seismic sensor, and formed the closed loop. The feedback mathematical model is shown in Equation (8).
(8)$$H_{feedback}=\frac{H_{seis}\left(s\right)H_{elec}\left(s\right)}{1+H_{seis}\left(s\right)H_{elec}\left(s\right)\cdot H_{f}\left(s\right)H_{coil}\left(s\right)}$$
where *H_elec_*(*s*) is the transfer function of the electrochemical seismic sensor, and *H_elec_*(*s*) = *H*_1_(*s*)**H*_2_(*s*)**H*_3_(*s*) is the signal adjusted circuits. The process of coil-magnet acted as *H_coil_*(*s*) = *k*_1_/*s*, in which *k*_1_ is the constant conversion coefficient from the voltage that loaded on the coil to feedback velocity, and the feedback control circuits *H_f_*(*s*) functioned as *k_f_***s*, to offset the influence of frequency, and *k_f_* was the depth of the feedback progress.

In Equation (8), if H_seis_(s)*H_elec_(s)*H_f_(s)*H_coil_(s) was much greater than 1, then Equation (8) could be rewritten as Equation (9), within the −3 dB bandwidth.
(9)$$H_{feedback}\approx \frac{1}{k_{1}k_{f}}$$

Thus, the sensitivities within the −3 dB bandwidth was related to the k_f_, and was controlled by the feedback control circuits.

### 2.3. Influences of Temperature on the Sensors

The influences of temperature on the electrochemical seismic sensor were mainly reflected in the following aspects, including ion diffusion coefficient, viscosity of the electrolyte, electrochemical reaction rate and elastic coefficient of the elastic membrane.

Convection and diffusion were two main sources of effective ions flux between the cathodes and the anodes. The flux caused by the diffusion of ions was J=−D∇C, where *D* is the diffusion coefficient of the ion and ∇C is the concentration gradient. When the concentration gradient is unchanged, the diffusion coefficient determined the values of the flux, and the expression of the diffusion coefficient could be expressed as [21]:(10)$$D=\frac{kT}{6\pi r\eta }$$

In Equation (10), *r* is the effective radius of the ion, η is the viscosity coefficient of the electrolyte, *k* is the Boltzmann constant, and *T* is the temperature of the electrolyte. Therefore, the diffusion process is directly affected by temperature.

Additionally, the viscosity (η in Equation (10)) of the electrolyte was also greatly affected by temperature. It is known from the literature [13] that the change of electrolyte viscosity had an exponential relationship with temperatures, and the change in flow resistance caused by viscosity affected the characteristic frequencies, and the higher temperature led to a smaller liquid damping force, which increased the sensitivity of the vibration process (see Figure 3). Generally speaking, the higher the temperature, the more convenient were the ion movement in the electrolyte, and the higher was the sensitivity of the sensor, but the natural convection noise increased accordingly [22].

The electrochemical reaction rate was also affected by temperature. As the temperature increased, the collision probability between the ions and the electrode increased greatly, which sped up the reaction rate and improved the sensitivity of the sensor.

The elastic membrane played an important role in the sensor. On the one hand, it provided a restoring force to the liquid mass, and on the other hand, it provided a feedback force to the system. When the temperature increases, the elastic membranes become soft, where *K* decreases, and the characteristic frequency shifts to the left. When the temperature decreases, the characteristic frequency shifts to the right. Therefore, the effect of temperature on the elastic membrane directly affects the shape of the sensitivity curves.

Therefore, under various temperature, the change of characteristic frequencies and the sensitivity of the amplitude-frequency curves should consider the influences of elastic coefficient, flow resistance, and electrochemical process.

## 3. Influence of the Surrounding Temperature on the Amplitude–Frequency Curves

In order to explore the influence of the surrounding temperature on the sensitivities of electrochemical seismic sensors, the open-loop and closed-loop sensitivity curves at different temperatures were tested. 

The −3 dB bandwidth of the sensor was adjusted to 0.01 Hz ~ 40 Hz in advance, through the feedback based on electromagnetic forces, and the sensitivity within the bandwidth was 2000 V/(m/s). The sensitive electrodes of the tested electrochemical seismic sensor were fabricated by the MEMS technologies, which had excellent performances in sensitivity and consistency [12].

### 3.1. Test Method for Temperature Sensitivities in Laboratory 

A dedicated test platform was built to test the sensitivity curves of the electrochemical seismic sensor at different temperatures. As shown in Figure 4a, the sensor was placed on a self-made support in a high-low temperature chamber (SH241, ESPEC, Tokyo, Japan, see Figure 4b).

Compared to Figure 1a, the seismic sensor was placed in a horizontal direction, the main difference between the horizontal and vertical was the effect of gravity. An additional support spring at the bottom was necessary for the sensors in vertical to prevent liquid mass from shifting too far from the center position (see in Figure 1a), which might increase the elastic coefficient of the seismic sensor, and cause an attenuation of sensitivities in a low frequency domain [23]. It was verified that the use of softer spring (151 ± 8 N/m in this study) could suitably offset the influence of gravity, the sensitivity change between horizontal and vertical were not very obvious.

There were two coil-magnet modules in the system (see Figure 4a,c). The coil-magnet on the right was used for closed-loop feedback, as shown in Figure 1a, and the coil-magnet on the left was used as the driving device. 

The working principles of the driving coil-magnet were as follows. The signal generator produced sinusoidal voltage with specific frequency and amplitude, and then these voltage signals were loaded onto the coil (left side in Figure 4a). The coil was fixed at the bracket. Then, the magnet inside the coil is affected by the electromagnetic force. The magnet is connected to the membrane and drives the movement of liquid mass to produce the input vibration velocity.

In addition, it was equipped with a thermometer to detect the temperature in the chamber. Due to the hysteresis for liquid thermal conductivity, the chamber was kept at a constant temperature for 1 h, before starting the test.

### 3.2. Test Results of Open-Loop and Closed-Loop Sensitivity Curves of Electrochemical Seismic Sensors at Different Temperatures

Since the electrochemical seismic sensors used in this study relied on the electrolyte of I_2_ & KI (0.02 mol/L & 2 mol/L), the freezing point was quantified as −13 °C after test, which could meet the needs of general outdoor environments. (If lithium iodide was used as the electrolyte, the freezing point could be further lowered to about −35 °C.) Therefore, this article only tested the sensitivity curves in the temperature range of −10 °C ~ +40 °C in the frequency range of 0.01 Hz ~ 100 Hz.

During the tests of sensitivities, we input sinusoidal velocity signals that were produced by driving the coil-magnet to the seismic sensor, and obtained the output voltage and the sensitivity at this frequency, which was the amplitude of the output voltage compared to the input speed amplitude. 

For example, when the temperature was 10 °C, the input velocity amplitude was 0.07 mm/s with the frequency at 1 Hz. The time domain diagram of the seismic sensor for the open-loop output is shown in Figure 5. It could be calculated that the voltage amplitude was 0.114 V, and the sensitivity was quantified as 1630 V/(m/s) @1 Hz. The high frequency noise in Figure 5a was caused by the temperature chamber. Figure 5b shows the voltage power spectrum signal. It could be seen that the signal strength was significantly higher than the local noise at a frequency of 1 Hz. The unit of the PSD was V^2^/Hz, and the values of PSD were processed logarithmically with 10 × log10(PSD) in this figure, and the unit was dB.

According to this method, the sensitivity value for −10 °C ~ 40 °C, at a frequency of 0.01 Hz ~ 100 Hz were tested, and the amplitude–frequency curves under this temperature were obtained. Figure 6 shows the open-loop and closed-loop sensitivity curves of the sensor at different temperatures. In which the sensitivities *S* (unit: V/(m/s)) were processed logarithmically with 20 × log10(*S*) in this study, and the unit was dB.

From the open-loop sensitivity curves in Figure 6a, it can be seen that the sensitivities of the sensor increased with the increase of the surrounding temperature in a non-linear manner. The largest change of sensitivities near the intermediate frequency was about 25 dB @1 Hz. The characteristic frequencies showed a right shift trend. It could be explained that the elastic coefficient was not the main influencing factor, and the influences of viscosity and the electrochemical process were the leading factors.

The closed-loop sensitivities curves are shown in Figure 6b, where the trends of sensitivity changes were not obvious, especially in the high-frequency domain. More specifically, the sensitivity curve at −10 °C showed a trend of attenuation with frequency, which might be due to the low temperature that led to the increase of viscosity and the decrease of the electrochemical reaction rate. In this scenario, the elastic membrane became harder, resulting in a significant decrease in sensitivity. Meanwhile, the feedback effect was weaker, and thus the curve could not be flattened. At 40 °C, the sensitivity curve increased with frequency, and even showed a resonance point, which might be due to factors such as softening of the elastic membrane.

Therefore, the mathematical models were only deduced for the open-loop sensitivity curves that had obvious trends.

In order to further analyze the influence of temperature on open-loop sensitivities, the open-loop sensitivity curves were fitted through typical elements. As shown in Figure 7, the curve at 20 °C was taken as an example, and the segments of 40 dB/dec, 20 dB/dec, 0 dB/dec, −20 dB/dec, and −40 dB/dec were used for an approximation, and thus the mathematical model of the open-loop transfer function could be written as:(11)$$H\left(s\right)=\frac{A_{T}s^{2}}{\left(1+\frac{s}{\omega_{1T}}\right)\cdot \left(1+\frac{s}{\omega_{2T}}\right)\cdot \left(1+\frac{s}{\omega_{3T}}\right)\cdot \left(1+\frac{s}{\omega_{4T}}\right)}$$
where $A_{T}$ is the sensitivity coefficient related to temperature, and $\omega_{1T}$, $\omega_{2T}$, $\omega_{3T}$, $\omega_{4T}$ are the frequencies at the intersections of the approximation segments in Figure 7, that is, the pole frequencies of the open-loop transfer function. Through the least square method, the mathematical model was fitted to the actual test curve to obtain the parameters, and then the open-loop transfer function at 20 °C could be expressed as:(12)$$H_{20℃}\left(s\right)=\frac{2089s^{2}}{\left(1+\frac{s}{0.71}\right)\cdot \left(1+\frac{s}{3.44}\right)\cdot \left(1+\frac{s}{43.81}\right)\cdot \left(1+\frac{s}{434.29}\right)}$$

According to this method, the open-loop sensitivity curves covering the whole temperature range were fitted, where the pole-frequency distributions and sensitivity coefficients at different temperatures are shown in Table 2.

It can be seen from Table 2 that both the pole frequencies *f*_1*T*_ and *f*_2_*_T_* increased with the increase of temperature. The changes of the *f*_3*T*_ and *f*_4*T*_ had no obvious relationship with the temperature, and thus, the influences of *f*_3*T*_ and *f*_4*T*_ on subsequent compensation were not considered here. The sensitivity coefficient A_T_ also increased with the increase of temperature, but the change was not very obvious.

## 4. Design and Verification of Temperature Compensation

Considering that the trends of the closed-loop sensitivity curves were not obvious, this paper designed the temperature compensation circuits based on thermistors, to realize open-loop temperature compensation, and then performed the closed-loop feedback to realize the final purpose of real-time temperature compensation.

As shown in Figure 8, it is a schematic diagram of the sensor in the closed-loop system, where the temperature compensation element H_comp_(s) was located inside the loop, and the thermistors were used to construct the temperature compensation circuits.

The functions of the temperature compensation circuits were to offset the influence of temperature on the pole frequencies and sensitivity coefficients of the open-loop transfer function. The circuits are shown in Figure 9.

Figure 9a is the compensation circuit for the pole frequencies, through the thermistor and resistance-capacitance combination, where a temperature-dependent compensation circuit was obtained. The transfer function of the compensation circuit could be expressed as Equation (13).
(13)$$H_{\mathrm{comp}1}\left(s\right)=A_{\mathrm{comp}1}\cdot \frac{\left(1+\frac{s}{\omega_{R}}\right)}{\left(1+\frac{s}{\omega_{0}}\right)}=\frac{C_{0}}{C_{1}}\cdot \frac{\left\{1+s\cdot \left[\left(R_{T}+R_{s1}\right)||R_{p}+R_{s2}\right]*C_{1}\right\}}{1+sR_{0}C_{0}}$$
where $A_{\mathrm{comp}1}$ is the magnification of the compensation circuit, and is consistent with $C_{0}/C_{1}$, generally $C_{0}$ equal to $C_{1}$. $\omega_{R}$ is the temperature-related characteristic frequencies in the compensation circuit, which was used to cancel the temperature-related pole frequencies $\omega_{T}$ in the sensor open-loop transfer function (Equation (11)), and ensure that the value of $\omega_{R}$ and the value of $\omega_{T}$ were consistent. $\omega_{R}$ was composed of $1/\left[\left(R_{T}+R_{s1}\right)||R_{p}+R_{s2}\right]*C$, where $R_{T}$ is the thermistor (R_25_
_°C_ =100 KΩ, B = 3950), and $R_{T}=R_{25\;℃}\cdot \exp\left[B\left(\frac{1}{T}-\frac{1}{T_{0}}\right)\right]$, where $T_{0}$ is 273.15 + 25 °C. The symbol of || stands for resistors in parallel. $\omega_{0}$ is the pole frequency at room temperature that could replace the temperature-related pole frequencies in Equation (11), after the cancellation of $\omega_{R}$ and $\omega_{T}$,
is composed of $1/R_{0}C_{0}$, and the value is the pole frequency measured at 20 °C in this study (see in Table 2).

Similarly, the compensation circuit for the sensitivity coefficient $A_{T}$ is illustrated in Figure 9b. Since it had nothing to do with the frequency, no capacitance was involved. The principle was to combine the thermistor and the conventional resistance to obtain the temperature-related transfer function, as shown in Equation (14):(14)$$H_{comp2}\left(s\right)=\frac{A_{0}}{A_{R}}=\frac{\left[\left(R_{T}+R_{s1}\right)||R_{p}+R_{s2}\right]}{R_{0}}$$

In the above equation, $A_{0}$ was the sensitivity coefficient at room temperature, which was equal to the data measured at 20 °C, as shown in Table 2. $A_{R}$ should be consistent with $A_{T}$ derived from the open-loop transfer function, in Equation (11).

In order to obtain the circuit parameters of effective compensation, according to the values in Table 2, $\omega_{R}=\omega_{T}=\left[\left(R_{T}+R_{s1}\right)||R_{p}+R_{s2}\right]*C_{1}$ (in which $\omega_{T}=2\pi f_{T}$) and $A_{R}=A_{T}=\frac{R_{0}}{\left[\left(R_{T}+R_{s1}\right)||R_{p}+R_{s2}\right]}\cdot A_{0}$ were fitted with the least square method. Since the open loop sensitivity curves had two poles and one sensitivity coefficient, third-order compensation circuits were required, and the fitting parameters are shown in Table 3.

Figure 9c is the circuit board used in the experiment, where three thermistors were used to form the third-order temperature compensation circuits.

To verify the effectiveness of the compensation circuits in Table 3, the pole frequencies and sensitivity coefficient fitted by typical elements (see in Table 2) were compared with the characteristic frequency and sensitivity coefficient constructed by the thermistor, as shown in Figure 10. It could be seen that they demonstrated a high consistency, which indicated that the temperature compensation circuits could offset the sensitivity changes effectively caused by temperature.

Using the designed temperature compensation circuits, the compensation effects of the electrochemical seismic sensor were measured, and the open-loop and closed-loop sensitivity curves after temperature compensation are shown in Figure 11. It could be seen from the test results that after temperature compensation, the open-loop and closed-loop sensitivity curves were greatly improved, as compared to those before compensation (see in Figure 6), and the sensitivity changes were controlled within 2 dB. Therefore, the temperature compensation method for electrochemical seismic sensors proposed in this study, was feasible and the effect was remarkable.

## 5. Conclusions

This study investigated the temperature characteristics of the open-loop sensitivity curves to find a solution to the sensitivities of the MEMS-based electrochemical seismic sensors, which are greatly affected by temperature. We proposed a mathematical model based on typical elements and designed the temperature compensation method based on thermistors. The results showed that this method had a significant effect and realized real-time temperature compensation.

## Figures and Tables

**Figure 1 micromachines-12-00387-f001:**
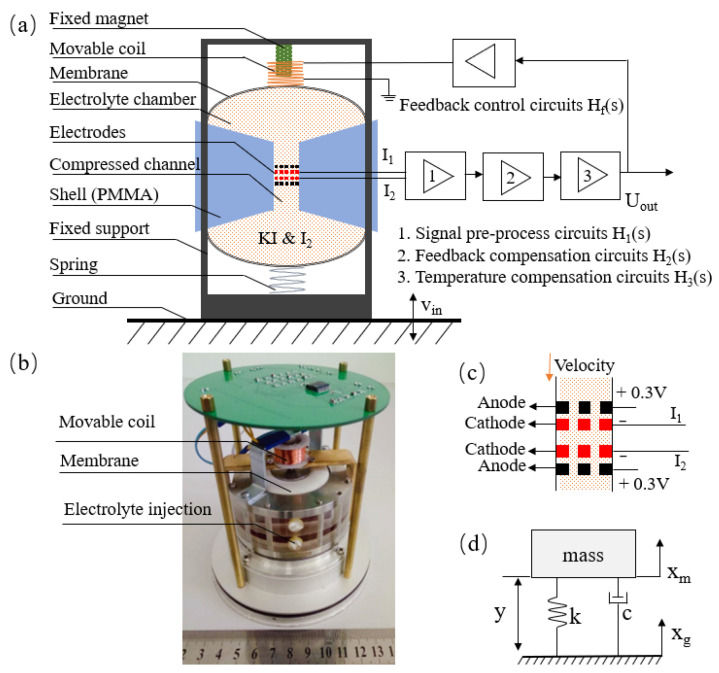
(**a**) The structure diagram of the electrochemical seismic sensor. The electrolyte that serves as the inertial mass transfer the outside vibration to the flow velocity, and then the change of ion concentration near the electrodes transfers to currents by the electrochemical reactions. (**b**) The prototypes of uniaxial electrochemical seismic sensors. (**c**) The sensitive electrodes of electrochemical seismic sensor, placed by ACCA at the center of compressed channel. (**d**) The diagram of the typical second-order inertial damping system.

**Figure 2 micromachines-12-00387-f002:**
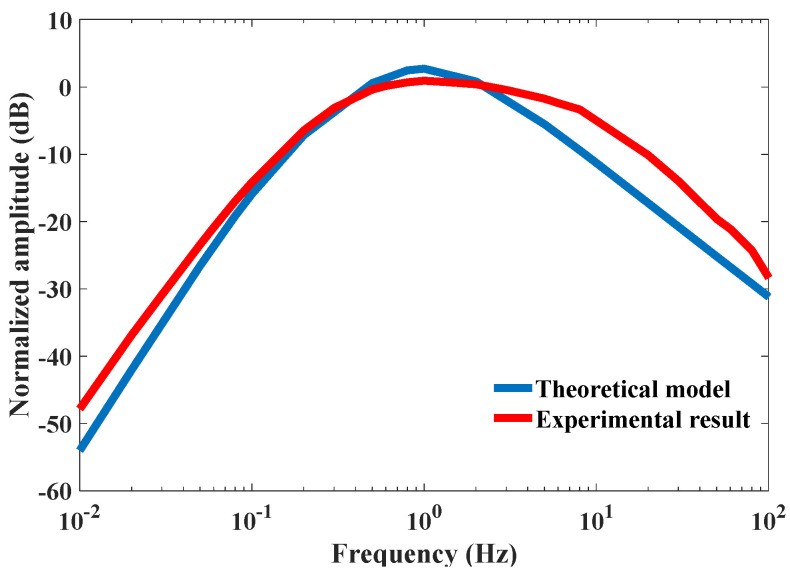
Transfer function of theoretical model and experimental result of the sensor.

**Figure 3 micromachines-12-00387-f003:**
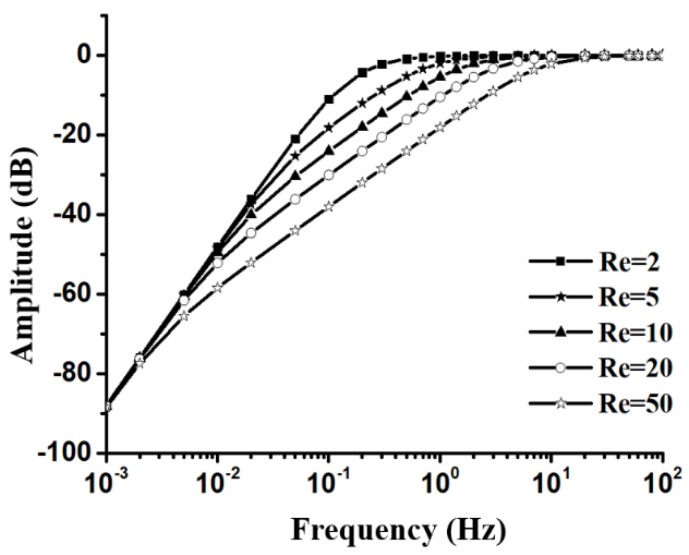
Vibration process with different equivalent flow resistances caused by the temperature-based viscosity, according to Equation (5).

**Figure 4 micromachines-12-00387-f004:**
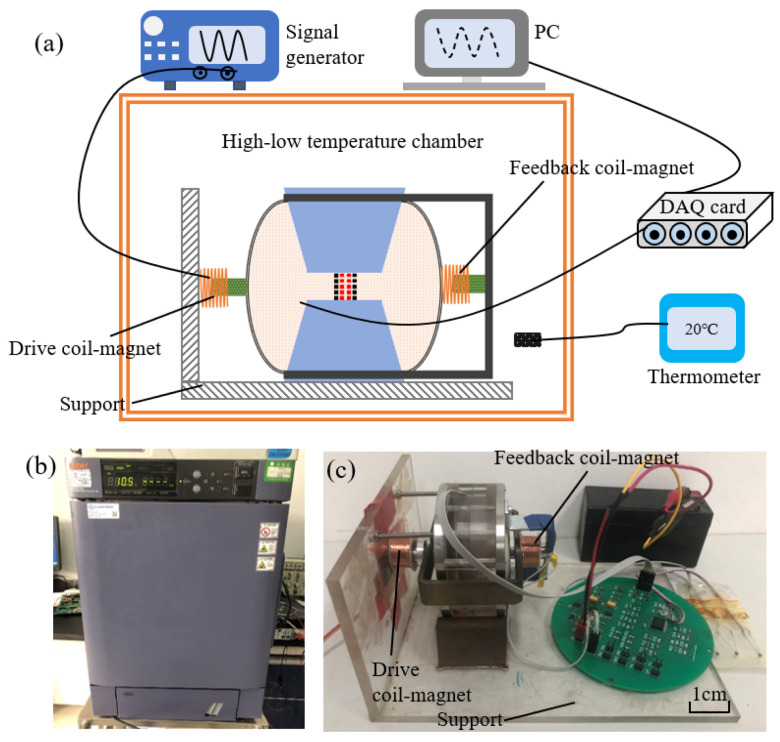
(**a**) The platform for temperature characteristic test, in which a high-low temperature chamber was used. (**b**) The image of the high-low temperature chamber. (**c**) The electrochemical seismic sensor with feedback coil-magnet and drive coil magnet, tested in the temperature chamber.

**Figure 5 micromachines-12-00387-f005:**
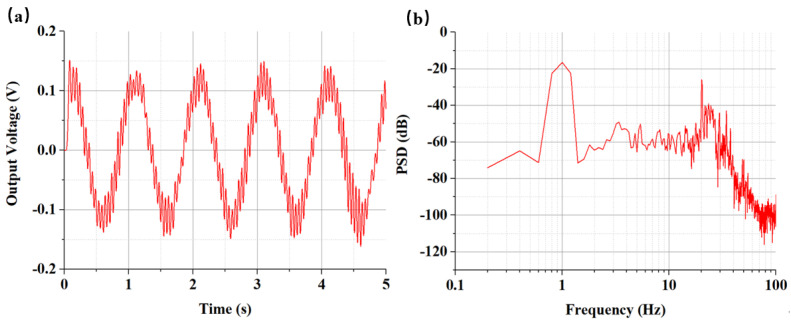
(**a**) The output voltage of the seismic sensor under the input velocity of 0.07 mm/s @1 Hz and with the temperature of 10 °C (**b**) The power spectrum signals of the test results of sensitivity at 1 Hz under the temperature of 10 °C.

**Figure 6 micromachines-12-00387-f006:**
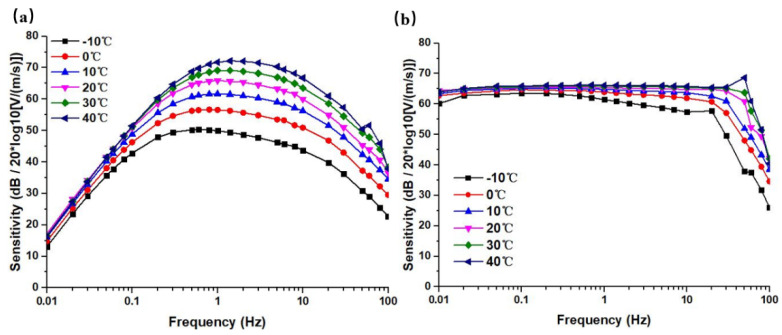
(**a**) The open-loop sensitivity curves at a temperature range of −10 °C ~ 40 °C. (**b**) The closed-loop sensitivity curves at a temperature range of −10 °C ~ 40 °C.

**Figure 7 micromachines-12-00387-f007:**
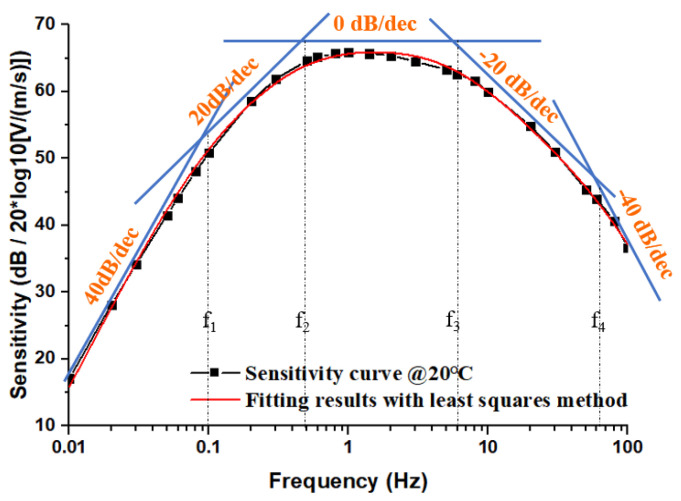
The method of fitting the open-loop transfer function.

**Figure 8 micromachines-12-00387-f008:**
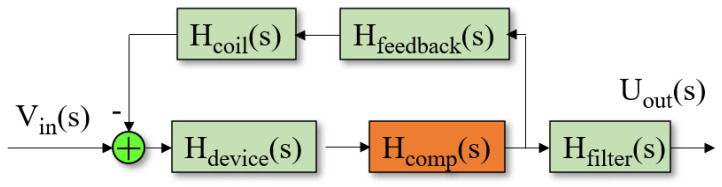
The flowchart for the closed-loop system with temperature compensation.

**Figure 9 micromachines-12-00387-f009:**
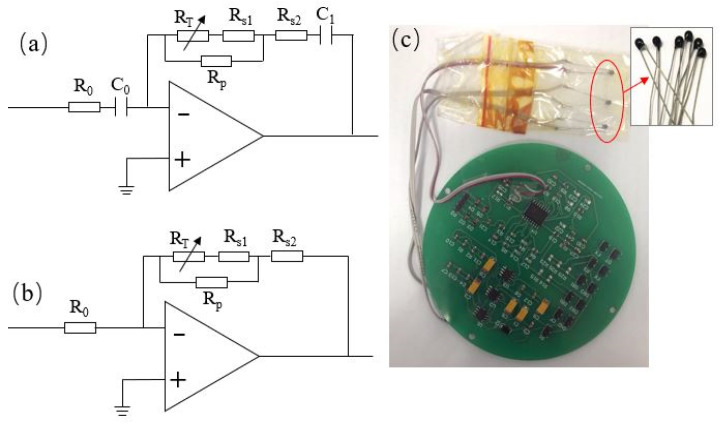
(**a**) The structure of compensation circuit for pole frequencies. (**b**) The structure of compensation circuit for sensitivity coefficient. (**c**) The image of the circuit board and thermistors.

**Figure 10 micromachines-12-00387-f010:**
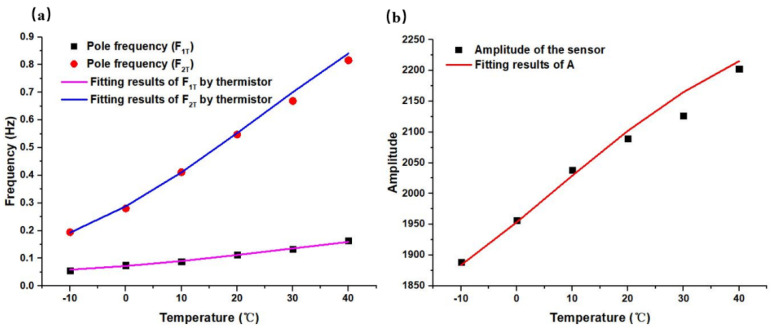
(**a**) Fitting results of pole frequencies. (**b**) Fitting results of sensitivity coefficient.

**Figure 11 micromachines-12-00387-f011:**
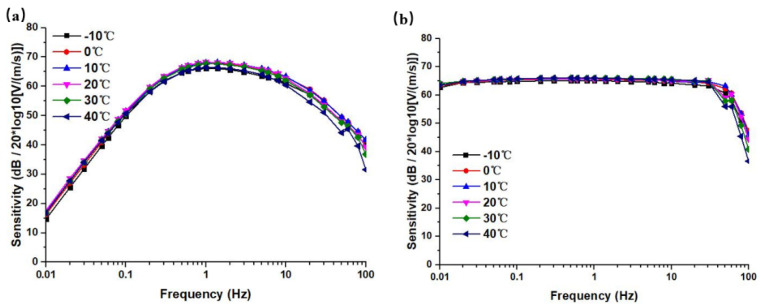
(**a**) Test results for open-loop sensitivity curves after temperature compensation. (**b**) Test results for closed-loop sensitivity curves after temperature compensation.

**Table 1 micromachines-12-00387-t001:** Property comparisons of different types of seismic sensors [12].

Working Principles	Company /Organization	Type	Bandwidth	Sensitivity V/(m/s)	Self-Noise @1 Hz	Power Consumption	Working Inclination
Electromagnetic	CGE Geological	CDJ-Z4	4 Hz–100 Hz	28	/	/	/
Capacitive	Streckeisen	STS-2.5	120 s–50 Hz	1500	−205 dB	450 mW	±0.48°
	Nanometrics	Trillium Compact	120 s–100 Hz	750	−190 dB	180 mW	±2.5°
	Guralp	3 T-360	360 s–50 Hz	1500	−206 dB	750 mW	±2.5°
Electrochemical	R-Sensors	CME6211	120 s–50 Hz	2000	−184 dB	360 mW	±15°
	PMD	BB603	120 s–50 Hz	2000	−185 dB	336 mW	±10°
	This work	ECS-100 s	100 s–40 Hz	2000	−175 dB	195 mW	±15°

**Table 2 micromachines-12-00387-t002:** The pole frequency distribution and sensitivity coefficients for open-loop sensitivity curves at different temperatures.

	−10 °C	0 °C	10 °C	20 °C	30 °C	40 °C
*f*_1*T*_ (Hz)	0.05599	0.0750	0.0884	0.1122	0.1341	0.1638
*f*_2*T*_ (Hz)	0.19431	0.2802	0.4112	0.5475	0.6688	0.8159
*f*_3*T*_ (Hz)	7.67711	6.82852	6.37973	6.97311	6.11246	6.54337
*f*_4*T*_ (Hz)	73.70842	64.81987	64.13743	69.15448	65.41345	69.1994
*A_T_*	1888.3	1956.4	2038.6	2089.0	2126.4	2202.5

**Table 3 micromachines-12-00387-t003:** Parameter fitting results of the temperature compensation circuits.

	*R* _0_	*C* _0_	*R_s_* _1_	*R_s_* _2_	*R_p_*	*C* _1_
Compensation circuit for $\omega_{1T}$	107 K	13.3 uF	39.6 K	10.3 K	313.3 K	13.3 uF
Compensation circuit for $\omega_{2T}$	154 K	1.9 uF	61.6 K	288 Ω	2.55 M	1.9 uF
Compensation circuit for $A_{T}$	121.3 K	/	108 K	59.6 K	86.1 K	/

## Data Availability

Not applicable.

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
