# Peer review of "Temperature Compensation of the MEMS-Based Electrochemical Seismic Sensors"

_micromachines, 2021, doi:10.3390/mi12040387_

Round 1

Reviewer 1 Report

Comments on micromachines-1149113

Built based on the authors’ previous works (e.g. Ref. 14 & 22), this manuscript characterize the temperature instability of the electrochemical seismic sensors, and proposed a temperature compensation circuits to improve the sensor performance in response to temperature variations. Overall, this is an interesting work. It’s worth to be published after addressing the following comments:

  • In the introduction, Line 33, why the electrochemical seismic sensors enable the benefits of 1) high sensitivity in low frequency domain, 2) no mechanical noises, and 3) large working inclination in compared with other mechanical seismic sensors? Please qualify this claim with data comparisons of item 1)-3) with other mechanical sensors (maybe list in a table). How is the sensitive of the seismic sensor defined? A vibration system as shown in Fig. 1C will be subjected to mechanical noises such as the thermomechanical noise. The electrochemical transduction will also be subjected to Brownian motion.

  • In the principles, section 2.1, transfer functions are given in eq. (2), (3) and (4). Are these theoretical transfer functions match with the experiment results? Please add the comparisons between the theoretical prediction and experimental data (e.g. data in fig. 3).

  • Line 119, why K is the Young’s modulus of membranes? From fig. 1a, K of the vibration system should be the effective spring constant, constituted by the serious connection of three springs: 1) the movable coil, 2) the membrane, and 3) the spring connecting the membrane and the fixed support.

  • Line 158, it is written that “when the temperature increases, the elastic membranes become soft, where K decreases, and the characteristic frequency shifts to the left. When the temperature decreases, the characteristic frequency shifts to the right.” As shown in fig. 3, the resonance frequency increases as the increase of temperature, which is an opposite phenomenon. Please explain.

  • Fig. 2a, why the sensor was aligned horizontally in the temperature chamber? Does the sensor behave similarly when it’s aligned in various directions, e.g. vertically or horizontally?

  • In the sensitive curves, e.g. fig.3, the unit of sensitivity is given as dB/20*log10 [V/(m/s)]. What does this unit mean? Please clarify and redefine the unit. How was the velocity (m/s) measured experimentally?

  • Line 166, please provide more details on why and how the 3 dB bandwidth of the sensor was adjusted. In section 3.2, please provide more details on how the close-loop feedback was performed for better readability of this manuscript.

  • In Fig. 8 and Fig.9, what is the mathematic model used for the curve fitting?

Reviewer 2 Report

The manuscript, “Temperature compensation of the MEMS based electrochemical seismic sensors” by Chao Xu, et al., describes the temperature compensation method using a thermistor-based feedback circuit to reduce the sensitivity spreads. The sensitivity variation due to temperature influence was greatly reduced over the measurement temperature range. However, the manuscript does not provide enough description how the sensitivity is defined and measured. It is recommended to add more experimental setup information and detailed experimental original data. Thus, a major revision is recommended. Below are my comments:

  1. As I mentioned above, there is no original response data in time domain. It is hard to understand the performance of the overall system under different frequencies/temperatures as well as the noise floor. Please add the original data in time domain as well as the calculation formula about the sensitivity calculation.
  2. The transfer function of the seismic sensor in equ. (2) (3) and (4) described the relationship between the system displacement and the overall electrochemical signal. Will the sensitivity have the same transfer function as well?
  3. In line 21-22, a couple questions:

1) How is the temperature range defined? The low limit is defined by the solution property but how about the high limit.

2) “…with the 3dB bandwidth…”, what’s the meaning of 3dB bandwidth? There is no detailed explanation about how 3dB bandwidth is defined.

3) “…0.01Hz~40Hz…”, how is the frequency range defined? From the characterization curve, the frequency goes up to 100 Hz, why only 40 Hz is stated at the introduction?

4). “…25dB…”, is it possible to characterize how much contribution of the variation from the electrochemical sensor part and the membrane?

  1. Line 59~62, is there any reason that why this sensor can go beyond “0.1 Hz” compared with the reference?
  2. Fig. 1(c), there is no detailed explanation about Fig. 1(c) in the context.
  3. Equ. (3), “~” is not a proper expression. Does it mean “equal to” or “proportional to” or something else?
  4. Line 137 ~ 162, it describes two key parameters to change the sensitivity, “K” and “D”. What’s the model about how sensitivity change vs. these two parameters under different temperature?
  5. Fig. 1 & 2, the sensor placement is rotated. Will the position of the sensor influence the measurement results by gravity?
  6. Fig. 1 & 2, what’s the function for the spring on the top and the spring on the other end? It is recommended a real picture of the sensor setup is shown in the context or in the supplementary instead of just showing an oven in Fig. 2(b).
  7. Fig. 3, there is an increase of the sensitivity at 40 C at higher frequencies, any explanation on this?
  8. Please provide the detailed feedback process in Fig. 1(a). It is not clear how the feedback was done and why there is such a difference between Fig. 3 & 4.
  9. Fig. 7, please demonstrate how the thermistor was mounted on the system. Is there any difference between the system temperature and the environment temperature?
  10. Fig. 10 & 11, after compensation, the sensitivity at 40C becomes smaller than other temperatures. Any explanations on this result?
  11. Please add the error bar and number of measurements for all the relevant plots.

Round 2

Reviewer 1 Report

The manuscript has been improved dramatically according to the comments. However, some of the previous comments were not fully addressed.

Comments (1):

Even relying on electrochemical process for transduction, the electrochemical seismic sensor is still subjected to thermomechanical noise. The seismic motion is detected as an output voltage signal by the following two steps 1) seismic-induced motion of the spring-mass vibrational system as shown in figure 1d, and 2) motion transduction of the mechanical vibrational system into electric signal by electrochemical process. The spring-mass vibrational system is subjected to thermomechanical motion noise.

Comments (2):

The authors mentioned that the trend of the theoretical models is consistent with the measurement. How consistent it is?

Comments (3):

What are the spring constants of the supporting spring and the movable coil respectively? Please mention in the manuscript that the movable coil has less contribution to the spring constants.

Comment (6):

Please provide more details (manufacturer and type) on the “standard” velocity sensor used for the measurement of the input velocity. The input velocity of the driving coil-magnet was calibrated on the vertical vibrational table. Does this calibration still hold when the driving coil-magnet was used horizontally?

In the new added figure 4, what’s the origin of the 20 Hz response when the device was driven under 1 Hz input velocity? Did the 20 Hz response show up consistently in other frequency measurements?    

Reviewer 2 Report

The author has addressed all my questions. The manuscript is recommended for publication.

Author Response

Thank you very much for your time.

Round 3

Reviewer 1 Report

Thanks for addressing my comments!